# Transplacental Transfer of Maternal Antibody against SARS-CoV-2 and Its Influencing Factors: A Review

**DOI:** 10.3390/vaccines10071083

**Published:** 2022-07-06

**Authors:** Shuang Liu, Jiayi Zhong, Dingmei Zhang

**Affiliations:** Department of Epidemiology, School of Public Health, Sun Yat-Sen University, Guangzhou 510080, China; liush333@mail2.sysu.edu.cn (S.L.); zhongjy58@mail2.sysu.edu.cn (J.Z.)

**Keywords:** COVID-19 vaccine, pregnancy, maternal antibody, antibodies transfer

## Abstract

Since the beginning of the coronavirus disease 2019 (COVID-19) outbreak, the disease has rapidly become a global threat. The constant emergence of new variants has increased the difficulty of controlling this disease. Vaccination is still considered the most effective method to prevent COVID-19. Vaccination has expanded to include children aged 3–17 years old, and some countries have lowered the age of vaccination to 6 months (for example, the United States). However, children under 3 years old are still not able to be vaccinated in most countries. In this study, we summarize the COVID-19 vaccination status in pregnant women, comprehensively elaborate on the status of maternal immune response and maternal antibody transfer after severe acute respiratory syndrome coronavirus 2 (SARS-CoV-2) infection and vaccination, and further analyze the possible influencing factors of maternal antibody transfer according to the currently available evidence on the topic. It was concluded that pregnant women develop an immune response and produce antibodies that can be transmitted through the placenta after vaccination, but more data are needed to determine the transfer rate and duration of these maternal antibodies and potential factors. The results provide a scientific basis for studying the protective effect of maternal antibodies on infants, formulating a vaccination strategy for pregnant women, and preventing SARS-CoV-2 infection in infants.

## 1. Introduction

Coronavirus disease 2019 (COVID-19), which is caused by severe acute respiratory syndrome coronavirus 2 (SARS-CoV-2), continues to spread worldwide and poses a serious threat to human health. According to the World Health Organization (WHO), as of 30 March 2022, a total of 483,556,595 confirmed cases have been reported worldwide, including 6,132,461 deaths [1]. The rapid spread of SARS-CoV-2 has brought many difficulties to effective prevention and control of the epidemic. At present, due to a lack of effective clinical treatments, the most effective method to end the COVID-19 epidemic is to establish population immunity.

Since the COVID-19 outbreak began, different kinds of COVID-19 vaccines have been quickly approved for marketing, including live-attenuated vaccines, inactivated vaccines, recombinant vector vaccines, subunit vaccines, and nucleic acid vaccines [2]. To date, approximately 11.2 billion doses of vaccine have been administered worldwide [1]. The data resulting from large-scale vaccination have confirmed the safety and efficacy of the available COVID-19 vaccines. For example, Thomas et al. [3] found that although vaccine efficacy decreases over time, vaccination is still safe and plays an essential role in the prevention of COVID-19. For COVID-19 variants, vaccination still provides immune protection. A real-world study found that the protection rate of inactivated vaccines against delta variant infection was 59.0%, and for moderate and severe COVID-19, the protection rates were 70.2% and 100%, respectively [4]. In addition, Pfizer announced that its phase 3 clinical trial data showed that the efficacy of the boosted Pfizer vaccine against the delta variant was as high as 95.6% [5].

As a special population, pregnant women and infants have received ongoing attention during the COVID-19 epidemic. Pregnant women have been excluded from vaccination clinical trials and the preliminary stages of large-scale vaccination, so data are currently lacking on the efficacy and safety of vaccination in this group. Due to physiological changes in the immune and cardiopulmonary systems, pregnant women are associated with a higher risk of serious COVID-19 infection and adverse pregnancy outcomes. Adequate vaccine safety and efficacy data have increased public confidence in the vaccine, and many countries have started to recommend that pregnant women receive the COVID-19 vaccine. Increasing lines of evidence indicate that vaccinating pregnant women with COVID-19 before and during pregnancy is safe and effective. Inoculation with a COVID-19 vaccine during pregnancy can not only stimulate an immune response in pregnant women, reducing the risk of infection and severe illness, but also generate antibodies that can be transferred from the mother to the fetus through the placenta, which may be of great significance for the protection of the mother and the newborn [6,7].

An infant’s immune system is not yet completely developed, and its ability to defend the body from the virus is weak. After infants suffer from COVID-19, the most common symptoms are respiratory symptoms, including respiratory distress, lack of oxygen, low oxygen saturation, and cough, and 38% of previously infected neonates require intensive care [8]. In previous studies on influenza vaccines, the maternal antibodies resulting from influenza vaccines have shown a clear protective effect on infants [9]. Thus, the WHO recommends that pregnant women receive influenza vaccines to prevent the flu.

Although studies have pointed out that pregnant women can transfer SARS-CoV-2 antibodies to the fetus after vaccination, helping the fetus to have some resistance to the virus, research conducted with a large enough sample size is lacking. The transfer rate and attenuation law of maternal antibodies as well as the factors and mechanisms affecting antibody transfer are still uncertain. In the present study, from the perspective of the vaccination status of children and pregnant women, we analyzed the antibody production and metastasis of pregnant women diagnosed with COVID-19 or who have been vaccinated with a COVID-19 vaccine during pregnancy. This work aimed to comprehensively describe the transmission efficiency, attenuation law, and influencing factors of maternal antibodies to provide a basis for formulating COVID-19 vaccination strategies for pregnant women and infants.

## 2. COVID-19 Vaccination of Pregnant Women Worldwide

The vaccination coverage of pregnant women is closely related to vaccination recommendations and the willingness of pregnant women to receive vaccination in each country; recommendations for vaccination during pregnancy also change over time [10]. Increasing amounts of research have indicated no obvious adverse reactions after pregnant women are vaccinated with a COVID-19 vaccine [11,12]. This finding provides evidence that vaccination is safe for pregnant women. Some countries have begun to encourage pregnant women to become vaccinated. For example, the British guidelines changed from the initial recommendation not to receive a COVID-19 vaccine during pregnancy to encouraging pregnant women to be vaccinated [13]. The US Centers for Disease Control and Prevention (CDC) also recommends that pregnant and breastfeeding women should receive COVID-19 vaccines [14]. Since May 2021, many countries in Asia have begun to recommend that pregnant women be vaccinated. Malaysia has suggested that vaccination be undertaken after 14 to 33 weeks of pregnancy [15]. Thailand and Singapore recommend every pregnant woman be vaccinated against COVID-19 after the 12th week of pregnancy [16,17]. However, China is more conservative and will not recommend vaccination for pregnant women until more data on the safety of vaccination for this population are available.

Although many national policies encourage pregnant women to become vaccinated, the willingness of pregnant women for vaccination is another important factor that affects vaccination. In a British survey [18], pregnant women’s willingness to be vaccinated against COVID-19 was significantly lower (69.9%) than that of non-pregnant women, mainly because they were concerned about whether vaccination would harm themselves or their babies. As a result, the vaccination coverage for pregnant women is low. One study surveyed 135,968 pregnant women between 14 December 2020 and 8 May 2021; 16.3% received ≥1 dose of a vaccine during their pregnancy, 5.3% started vaccination during their pregnancy, and 11.1% were vaccinated completed [19]. In a recent national prospective cohort study in Scotland, the vaccination rate of pregnant women was lower than that of the general female population aged 18–44 years old. In October 2021, 32.3% of women who gave birth received two doses of vaccine compared with 77.4% of all women. Only 0.9% of pregnant women received a third or enhanced vaccination [20]. It should be noted that the willingness of pregnant women to receive vaccinations is closely related to the advice provided by their health care providers. Previous studies on influenza vaccines, pertussis vaccines, and Tdap vaccines have found that the advice of health care providers is the most important factor in improving vaccination coverage [21,22]. Thus, the advice of health care providers may also help increase COVID-19 vaccination coverage in pregnant women.

Pregnant women who are infected with SARS-CoV-2 are at a high risk of serious maternal and neonatal complications. In a meta-analysis covering 61 studies, 790 COVID-19-positive women and 548 newborns were analyzed. The incidence rates of cesarean section, premature delivery, low birth weight, and adverse pregnancy events were 72%, 23%, 7%, and 27%, respectively [23]. Compared with women who had not been diagnosed with COVID-19, the women who had been diagnosed with COVID-19 had a significantly increased risk of serious pregnancy complications, including preeclampsia, eclampsia, HELLP (hemolysis, elevated liver enzymes, and low platelets) syndrome, ICU hospitalization, premature delivery, and low birth weight [24]. In a systematic review, out of 1,214 children under 5 years old who were confirmed to be infected with COVID-19, 51% were infants [25]. Young children, especially infants, are vulnerable to infection, especially when new variants appear, which may cause serious health concerns [26].

Many countries around the world have begun to vaccinate children aged 3–17 years old against COVID-19. For example, China expanded the vaccination age to three years old and above in November 2021 [27]. In Colombia, Argentina, and other countries using the inactivated COVID-19 vaccine, children over the age of 3 years of age have begun to be vaccinated [28,29]. In Europe, after COVID-19 vaccines had been approved for those aged 12 and above, the European Medicines Agency (EMA) further recommended COVID-19 vaccinations for children aged 5 to 11 [30]. In the United States, the vaccination recommendations for children were updated in June 2022, from initially offering vaccines to children aged 5 to 11 years to those over six months old [31]. Nevertheless, the vaccination rate of children is still lower than that of other age groups, and infants under 3 years old are still unable to be vaccinated in most countries. As a result, achieving immunization in children is critical, especially in newborns with an immature immune system, and vaccinating their mothers is the best strategy that is currently available. According to the current vaccination situation, the vaccination rate of pregnant women and children is relatively low, and children under 3 years old are not eligible to be vaccinated in most areas. During the COVID-19 pandemic, immune protection should be established for special groups, such as pregnant women and children. Therefore, we need more data on vaccine safety and efficacy to create a scientific basis for developing vaccination strategies for pregnant women and children.

## 3. Maternal Antibody Transfer after SARS-CoV-2 Infection

In a systematic review of 49 studies including a total of 655 pregnant women and 666 neonates, no clear evidence of vertical mother-to-child COVID-19 transmission was found [32], although perinatal transmission was suspected in a small number of cases [33,34]. At present, some studies have found that pregnant women have the same specific immune response as ordinary people after being infected with SARS-CoV-2.

To determine whether the antibodies produced by the mother after being infected with SARS-CoV-2 can be effectively passed to a newborn during pregnancy, a cohort study in the United States detected the IgG antibody in 83 maternal and 72 cord blood sera. The antibody-negative neonates were born to pregnant women who had IgM-only or lower IgG antibody concentrations. These pregnant women with SARS-CoV-2 infection had a placental antibody transfer rate of more than 1.0 [35]. In addition, a study in China including 26 pregnant women and 27 newborns reported that 80.8% of mothers and half of their newborns acquired antibodies against SARS-CoV-2 after delivery. Moreover, the seroconversion rate in symptomatic mothers before delivery was significantly higher than that in asymptomatic mothers [36]. Similarly, another prospective cohort study showed that the IgG of the anti-S protein receptor-binding domain (RBD) was detected in 91% of umbilical cord blood samples from pregnant women infected with SARS-CoV-2. The median cord/maternal antibody ratio was 0.81, which was lower than expected [37]. In addition, two studies from China and Italy similarly reported that antibodies could be passed through the placenta to the newborn after a mother was infected with COVID-19 [38,39]. An American study had similar conclusions and found that the antibody titers and functional activity in the umbilical cord were lower than those in maternal plasma [40]. Rathberger et al. [41] further reported that in mothers with high antibody titers at birth, the antibodies could be transferred through breast milk as well.

As SARS-CoV-2 maternal antibodies can be effectively delivered to neonates, another scientific concern is how long they will last. Wang X et al. demonstrated that the SARS-CoV-2 IgG levels of newborns with SARS-CoV-2-infected mothers dropped sharply to one-tenth two months after birth [36]. Similar results were reported in a prospective study from Germany [41] that included 16 SARS-CoV-2-infected pregnant women and their newborns. In Rathberger’s study, antibodies against SARS-CoV-2 were detected, but the antibody titers decreased or disappeared in infants at follow-up 6–11 weeks after birth. Meanwhile, maternal antibody levels remained stable and were even partially enhanced. Additionally, Joseph et al. [37] stated that only 25% of the umbilical cord blood samples with detectable IgG against the RBD of the S protein had a neutralizing ability compared with 94% in maternal blood.

SARS-CoV-2 antibodies can indeed be transmitted to newborns through the maternal placenta (as shown in Table 1), but the transfer rate is lower than expected, and the neutralization ability is significantly decreased. The antibody level in newborns gradually declines or disappears over time [35,37,41]. Regarding factors affecting the transfer rate and duration of maternal antibodies against COVID-19, Flannery et al. [35] affirmed that the concentration of IgG in cord blood was positively correlated with the concentration of maternal IgG (r = 0.886, *p* < 0.001); this conclusion was also confirmed by Joseph and Rathberger et al. [37,41]. However, whether the length of the interval between the mother’s infection with SARS-CoV-2 and childbirth is related to the transfer rate of maternal antibodies remains unclear. Flannery and Rathberger et al. [35,41] found a correlation between them, but Joseph [37] came to the opposite conclusion. In addition, another study focused on factors such as FcRn expression, IgG subclass, antigen structure, and IgG glycosylation, which may be related to the specific transfer of SARS-CoV-2 antibodies [40]. Many studies have also focused on exploring the influencing factors or mechanisms of these critical issues to provide new directions for further research (Table 1).

## 4. Maternal Antibody Transfer after SARS-CoV-2 Vaccination

Studies have confirmed not only that pregnant women infected with SARS-CoV-2 produce specific antibodies but also that these antibodies can be transferred to the fetus through the placenta, which may have a protective effect on infants. However, additional data are needed to clarify whether the immune response and antibody transfer following COVID-19 vaccination during pregnancy are similar to those in pregnant women infected with SARS-CoV-2.

Israel was the first country to start vaccinations and boost immunization. Similarly, Israel was one of the first countries to recommend vaccination for pregnant women, providing us with a lot of vaccination experience. A study in Israel found that the BNT162b2 COVID-19 vaccine induced strong maternal immunization during pregnancy, and the antibodies were effectively transferred to the fetus, supporting the effects of vaccination during pregnancy [42]. Another Israeli study compared pregnant women vaccinated with a COVID-19 vaccine with pregnant women infected with SARS-CoV-2; in the comparison, 100% maternal and 98.3% cord blood serum samples were positive for SARS-CoV-2 IgG, illustrating that infection with SARS-CoV-2 and vaccination can induce strong maternal immunity during pregnancy and that IgG can be effectively transferred [43]. As shown in Table 2, studies from the United States, Poland, and other countries have obtained similar results through the follow-up of vaccinated pregnant women [44,45,46]. In addition to the placental pathway, antibodies were found to be present in the milk of pregnant women injected with an mRNA COVID-19 vaccine [47], suggesting that immune transfer to the newborn might also be possible through breast milk.

The maternal antibody level in neonates and its persistence affect the protection and duration of antibodies during infancy. In a cohort of newborns born to women vaccinated against SARS-CoV-2, the titer of IgG antibodies in the cord blood was significantly higher than it was in those born to infected individuals [48]. The follow-up of infants born to two categories of pregnant women revealed that most infants born to mothers who had received COVID-19 vaccination still had persistent antibodies after six months [48]. Compared with pregnant women infected with SARS-CoV-2, the maternal antibody titers transmitted to the fetus by vaccinated pregnant women were present at higher levels and were longer-lasting. Similarly, the placental transfer of antibodies plays an important role in maternal antibody titers in neonates, with an average IgG transfer rate of 1.0 ± 0.6 in 27 pregnant women vaccinated during pregnancy, which is slightly lower than the reported pertussis vaccination IgG transfer rate (1.19–1.36) [49]. In Zdanowski et al.’s study [42], the average SARS-CoV-2 antibody transfer rate was 1.28 ± 0.798. Therefore, we need additional data to determine the placental transfer rate of the SARS-CoV-2 antibody. The factors and mechanisms that affect the rate and amount of placental transfer of maternal antibodies should also be investigated, as this information is of great significance for improving the rate of maternal antibody transfer and protecting infants from COVID-19 infection.

First, whether or not the mother has been completed vaccinated is a factor. A study showed that only 43.6% of newborns born to women who received one dose of vaccine had detectable IgG, while 98.5% of newborns born to women who received two doses of vaccine had detectable IgG [48]. Moreover, the IgG antibody levels in pregnant women who received booster vaccination were significantly higher than those in women who received two doses of the vaccine [49]. Similar to pregnant women infected with COVID-19, the maternal serum from vaccinated pregnant women had a positive correlation with antibody titers in cord blood [7,50]. In addition, studies have found that the placental transfer rate is related to how many weeks remain until delivery when pregnant women received their first or second doses of vaccine. The maternal IgG levels and placental IgG transfer rate increase over time, indicating that the time interval between vaccination and delivery may be an important factor affecting the placental transfer of antibodies [51,52,53,54]. Therefore, to promote antibody transfer through the placenta, we need to determine the best time to vaccinate pregnant women. Studies have shown that in the early third trimester (the first dose at 27–31 weeks, median: 9620 AU/mL), antibody transfer through the placenta is higher than it is in the second trimester (the first dose at 19–26 weeks, 3970 AU/mL) and in the late third trimester (the first dose at 32–36 weeks, 6697 AU/mL) [53]. However, other studies suggest that maternal vaccination from the early second trimester of pregnancy onward may be the best choice to achieve the highest level of antibodies in newborns [55]. At present, the optimal vaccination time for pregnant women is still uncertain, and a prospective follow-up study with a larger sample size is needed to determine it.

Vaccination is essential for protecting pregnant women and their offspring from SARS-CoV-2 infection. Studies have shown that vaccinated women have significantly reduced COVID-19 infection during pregnancy compared with unvaccinated women [56] and that vaccination provides partial protection for pregnant women and their offspring, even in the face of some variants [57]. However, additional data are needed to determine how much these antibodies can protect newborns. Using influenza vaccines as an example, the protection rate of maternal antibodies in infants under six months of age after vaccination during pregnancy was 91.5% [58], demonstrating that the maternal antibodies resulting from the influenza vaccine had a clear protective effect on infants. In addition, the diphtheria and tetanus combined vaccine exerted a protective effect on infants administered to pregnant women [59]. Whether antibodies produced by maternal COVID-19 vaccination protect neonates in a similar fashion to other vaccines remains unknown. In one study, mRNA COVID-19 vaccination was 61% (95% CI = 31–78%) effective against infants younger than 6 months of age who needed to be hospitalized for COVID-19 if two doses had been administered to their mothers during pregnancy. Hence, receiving two doses of an mRNA COVID-19 vaccine during pregnancy may help prevent the hospitalization of infants under 6 months of age [60]. To consider maternal immunity for the protection of infants, we need data to understand the kinetics and duration of maternal antibodies in infants as well as their neutralizing activity and efficacy against COVID-19.

## 5. Conclusions

The COVID-19 epidemic has been ongoing for more than two years and has spread worldwide. Maternal immunization can not only protect the mother and fetus from SARS-CoV-2 infection but also provide passive protection for babies after birth due to the antibodies transmitted through the placenta. However, some problems persist: the best time for mothers to become vaccinated to protect newborns is unclear; the transfer rate of maternal antibodies and the duration and level of maternal antibody transfer in infants also require more data. The data needed to determine the extent to which maternal antibodies protect lactating newborns remain insufficient; whether these antibodies have potential adverse effects on the active immune response of newborns and how much they can help protect infants from SARS-CoV-2 infection, serious diseases, and death also remains unclear. We need to explore unknown mechanisms that may affect the antibody transfer process from mother to child. Therefore, against the background of the ongoing COVID-19 pandemic, information should be collected through well-designed prospective or longitudinal studies to provide a scientific basis for the safe and effective implementation of maternal and pediatric vaccination strategies.

## Figures and Tables

**Table 1 vaccines-10-01083-t001:** Summary of several studies on maternal and fetal antibodies after infection with severe acute respiratory syndrome coronavirus 2 (SARS-CoV-2).

Author	Region	Study Design	Detectable Antibody	Transfer Rate	Relevant Factors
Flannery et al. [35]	America	cohort study	mothers: 100% (83/83) infants: 87% (72/83)	1.02 (0.85–1.23)	infection and delivery interval; maternal IgG concentration
Atyeo et al. [40]	America	cohort study	N/A	significantly lower	FcRn level; IgG glycosylation; IgG subclass and antigen structure
Joseph et al. [37]	America	prospective cohort study	mothers: 100% (32/32) infants: 91% (29/32)	0.81 (0.67–0.88)	maternal IgG concentration
Wang et al. [36]	China	prospective cohort study	mothers: 80.8% (21/26) infants: 44.4% (12/27)	18.8% (<2 weeks) 81.8% (>2 weeks)	maternal IgG concentration; symptomatic or asymptomatic before delivery
Zeng et al. [38]	China	case report	mothers: 83.3% (5/6) infants: 83.3% (5/6)	N/A	N/A
Rathberger et al. [41]	Germany	prospective cohort study	mothers: 73.0% infants: 1/3	N/A	infection and delivery interval; maternal IgG concentration
Cavaliere et al. [39]	Italy	case report	mothers: 100.0% (1/1) infants: 100.0% (1/1)	N/A	N/A

N/A: not available.

**Table 2 vaccines-10-01083-t002:** Summary of several studies on maternal antibodies in pregnant women and fetuses after the coronavirus disease 2019 (COVID-19) vaccination.

Authors	Study Design	Region	Vaccine	Neonatal Antibody Positive Ratio	Antibody Transfer Rate	Vaccination Time	Factors
Soysal et al. [43]	Case report	Turkey	CoronaVac	100% (1/1)	1.04	First dose at 28 weeks	N/A
Zdanowski et al. [42]	Retrospective study	Poland	Pfizer-BioNTech	100% (16/16)	1.28 ± 0.798 (mean)	First dose at 29–36 weeks	The time from vaccination to delivery
Mithal et al. [47]	Prospective study	US	Moderna and Pfizer-BioNTech	89.3% (25/28)	1.0 ± 0.6 (mean)	First dose at 31~35 weeks	The time from vaccination to delivery
Kugelman et al. [54]	Prospective cohort study	Israel	Pfizer-BioNTech	100% (114/114)	2.6 (median)	First dose at 18.6~25.2 weeks	Mother’s age; the time from vaccination to delivery
Nir et al. [7]	Prospective cohort study	Israel	Pfizer-BioNTech	98.3% (63/64)	0.77 (median)	Completed vaccination 21.7 ± 11 days before delivery.	N/A
Kugelman et al. [44]	Prospective cohort study	Israel	Pfizer-BioNTech	100% (93/93)	1.4 (median)	Completed the third dose at 32.2 ± 3.2 weeks of gestation.	Duration from last vaccine to birth was demonstrated

N/A: not available.

## Data Availability

Not applicable.

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
