# Peer review of "Transplacental Transfer of Maternal Antibody against SARS-CoV-2 and Its Influencing Factors: A Review"

_vaccines, 2022, doi:10.3390/vaccines10071083_

Round 1

Reviewer 1 Report

The manuscript summarizes the current scientific evidencies available on uptake of maternal vaccination for COVID 19, rate and influencing factors of transplacental passage of maternal antibodies in vaccinated mothers compared to infected mothers. The aim is to provide strong basis to recommend maternal vaccination against COVID 19 to protect infants (for which no vaccine is currently available) from the disease.

The topic is interesting. However English editing and some changes are needed to make the manuscript suitable for publication.

Abstract: Clarify that the manuscript summarizes available evidences on the topic. At present is it not clear that the paper doesn’t report any additional research, therefore creating expectations that can’t be met by the manuscript.

Page 2, lines 50-52, Introduction: “After the virus infects the fetus, it can lead to abortion, intrauterine growth restriction, premature delivery, and other adverse pregnancy outcomes or may cause serious birth defects [6-7].’’: I was not aware that SARS CoV 2 might have teratogenic effects on the fetus. If this is the case, please cite the correct references (6 and 7 are manuscripts that describe kinetics of SARS CoV2 specific antibodies production).

Pages 2-3, lines 78-113, Paragraph 2: I find this paragraph to pertinent to the manuscript. I suggest to delete it, and summarize the concept with: “No vaccine is currently available for children under 3 years of age. To protect infants, maternal vaccination is therefore at the present moment the best strategy available” – that is actually already written at Page 4, Lines 157-163.

Page 3, line 144, Paragraph 3: I suggest to add a sentence about the fact that willingness of pregnant women to get vaccinated is strongly related to the recommendation from healthcare providers. This is well reported in literature ( e.g. Differences between influenza and pertussis vaccination uptake in pregnancy: a multi-center survey study in Italy, Vilca L.M., Eur J Public Health. 2021 ; Flu and Tdap Maternal Immunization Hesitancy in Times of COVID-19: An Italian Survey on Multiethnic Sample, Cavaliere A.F., Vaccines 2021). Therefore it is likely that the more consistent the guidelines on maternal vaccination against COVID-19 will become the higher will be the uptake of immunization in pregnant women.

Page 4, lines 165-169, Paragraph 4: I suggest to replace first data about COVID 19 vertical transmission with more recent and bigger studies (e.g. Walker KF, Maternal transmission of SARS-COV-2 to the neonate, and possible routes for such transmission: a systematic review and critical analysis. BJOG 2020).

Page 5, Table 1: this is not the summary of all studies about transplacental passage of maternal antibodies in natural infection, many others have been published (e.g. Zeng H, Antibodies in Infants Born to Mothers With COVID-19 Pneumonia. JAMA 2020; Cavaliere A.F., Passive immunity in newborn from SARS-CoV-2-infected mother. J Med Virol. 2021 just to cite two of them). The Title of the Table, “Summary of research status” is therefore misleading. I suggest to add all the manuscript published on the topic, or to change the Title.

Page 7, Table 2: this is not the summary of all studies about transplacental passage of maternal antibodies in vaccinated mothers, many others have been published. The title of the Table, “Summary of research status” is therefore misleading. I suggest adding all the manuscript published on the topic. I suggest to add all the manuscript published on the topic, or to change the Title.

Author Response

Dear Reviewer:

    Thank you for your comments concerning our manuscript entitled “Transplacental transfer of maternal antibody against SARS-CoV-2 and its influencing factors:a review”(Manuscript ID: vaccines-1716269). Those comments are all valuable and very helpful for revising and improving our paper, as well as the important guiding significance to our researches. We have studied comments carefully and made correction which we hope meet with approval. To make it easier for you to view the manuscript, we used the "Track Changes" function of MS Word and in red. The main corrections and responses to reviewers' comments in the paper  are attached(Response to Reviewer 1 Comments. docx).

     Once again, thank you very much for your comments and suggestions.

Yours sincerely,

Shuang Liu

Corresponding author: Dingmei Zhang

Affiliation: Department of Epidemiology, School of Public Health, Sun Yat-sen University, Guangzhou, Guangdong 510080, China

Response to Reviewer 1 Comments

Point 1: English editing is needed to make the manuscript suitable for publication.

Response 1: Thank you for your suggestions. According to your advice, this manuscript has been edited for proper English language, grammar, punctuation, spelling, and overall style by MDPI English editing service.

Point 2: Abstract: Clarify that the manuscript summarizes available evidences on the topic. At present is it not clear that the paper doesn’t report any additional research, therefore creating expectations that can’t be met by the manuscript.

Response 2: Thank you very much for your suggestion. We have added a sentence to the abstract to clarify this issue:"…according to the currently available evidence on the topic. It was concluded that pregnant women develop an immune response and produce antibodies that can be transmitted through the placenta after vaccination, but more data are needed to determine the transfer rate and duration of these maternal antibodies and potential factors.” (Page 1, Lines 17-20)

Point 3: Page 2, lines 50-52, Introduction: “After the virus infects the fetus, it can lead to abortion, intrauterine growth restriction, premature delivery, and other adverse pregnancy outcomes or may cause serious birth defects [6-7].’’: I was not aware that SARS CoV 2 might have teratogenic effects on the fetus. If this is the case, please cite the correct references (6 and 7 are manuscripts that describe kinetics of SARS CoV2 specific antibodies production).

Response 3: Thank you for pointing out this problem. Previously, as we reviewed some literature on the effects of viruses on mothers and their fetuses, we learned that certain viral infections in mothers are associated with fetal birth defects (such as Zika virus). In order to avoid misunderstanding, we have deleted this sentence. (Page 2, Lines 52-54)

Point 4: Pages 2-3, lines 78-113, Paragraph 2: I find this paragraph to pertinent to the manuscript. I suggest to delete it, and summarize the concept with: “No vaccine is currently available for children under 3 years of age. To protect infants, maternal vaccination is therefore at the present moment the best strategy available” – that is actually already written at Page 4, Lines 157-163.

Response 4: Thanks for this constructive suggestion. We deleted the second paragraph from the original manuscript, and change part of the content of children's vaccination to lines 168-176,which is used to explain that children under 3 years old are currently not covered by vaccination.

Point 5: Page 3, line 144, Paragraph 3: I suggest to add a sentence about the fact that willingness of pregnant women to get vaccinated is strongly related to the recommendation from healthcare providers. This is well reported in literature ( e.g. Differences between influenza and pertussis vaccination uptake in pregnancy: a multi-center survey study in Italy, Vilca L.M., Eur J Public Health. 2021 ; Flu and Tdap Maternal Immunization Hesitancy in Times of COVID-19: An Italian Survey on Multiethnic Sample, Cavaliere A.F., Vaccines 2021). Therefore it is likely that the more consistent the guidelines on maternal vaccination against COVID-19 will become the higher will be the uptake of immunization in pregnant women.

Response 5: We deeply appreciate the reviewer’s suggestion. According to the reviewer’s comment, we have added several sentences regarding the fact in lines 150-155, Paragraph2..

Point 6: Page 4, lines 165-169, Paragraph 4: I suggest to replace first data about COVID 19 vertical transmission with more recent and bigger studies (e.g. Walker KF, Maternal transmission of SARS-COV-2 to the neonate, and possible routes for such transmission: a systematic review and critical analysis. BJOG 2020).

Response 6: Thank you for your kind suggestions, which is valuable for improving the quality of the manuscript. We have replaced the previous one with the article you recommended (reference 28), to confirm our conclusions about vertical transmission of COVID-19 with a more recent and bigger study. At the same time, we have also revised the relevant statement according to the content of the newly cited literature (page 4, lines 184-187).

Point 7: Page 5, Table 1: this is not the summary of all studies about transplacental passage of maternal antibodies in natural infection, many others have been published (e.g. Zeng H, Antibodies in Infants Born to Mothers With COVID-19 Pneumonia. JAMA 2020; Cavaliere A.F., Passive immunity in newborn from SARS-CoV-2-infected mother. J Med Virol. 2021 just to cite two of them). The Title of the Table, “Summary of research status” is therefore misleading. I suggest to add all the manuscript published on the topic, or to change the Title.

Response 7: Thank you for your valuable suggestion. We added some papers and changed the title of the Table 1 to " Summary of Some Studies on Maternal and Fetal Antibodies after Infection with Severe Acute Respiratory Syndrome Coronavirus 2 (SARS-CoV-2)".

Point 8: Page 7, Table 2: this is not the summary of all studies about transplacental passage of maternal antibodies in vaccinated mothers, many others have been published. The title of the Table, “Summary of research status” is therefore misleading. I suggest to add all the manuscript published on the topic, or to change the Title.

Response 8: Thanks a lot for the reviewers' suggestion. We changed the title of the Table 2 to " Summary of Some Studies on Maternal Antibody in Pregnant Women and Fetuses after the Coronavirus Disease 2019 (COVID-19) Vaccination ". We hope the revised title will be more appropriate. To enrich the content, we also added a literature on maternal antibody transfer status after the third dose of inoculation(As shown in Table 2).

Reviewer 2 Report

The authors in the review “Transplacental transfer of maternal antibody against SARS-CoV-2 and its influencing factors:a review” report the COVID-19 vaccination status in children and pregnant women and analyse the maternal antibody transfer after SARS-CoV-2 infection and vaccination. The review is well organized and clearly writed. In the literature there are not many data on the subject and the focus of the review has a great and important social impact. The publication of the review by Marco A. P. Safadi on journal Am J Reprod Immunol. (April 22, 2022) reduces the novelty of the review which, however, remains important to enrich the literature; add the Safadi’s reference.

Finally, I suggest to cited the Ching-Ju Shen’ article to strengthen the concept that Immunization of pregnant women is crucial to protect mothers and their offspring from infection.

The review is interesting and should be published after undergoing the following minor corrections:

  • Add the review: Safadi MAP, Spinardi J, Swerdlow D, Srivastava A. “COVID-19 Disease and Vaccination in Pregnant and Lactating Women.” Am J Reprod Immunol. 2022 Apr 22. doi: 10.1111/aji.13550.
  • Add the article: Ching-Ju Shen , Yi-Chen Fu , Yen-Pin Lin , Ching-Fen Shen , Der-Ji Sun , Huan-Yun Chen. “Chao-Min Cheng. Evaluation of Transplacental Antibody Transfer in SARS-CoV-2-Immunized Pregnant Women”. 2022 Jan 10;10(1):101. doi: 10.3390/vaccines10010101.

Author Response

Dear Reviewer,

    Thank you for your comments concerning our manuscript entitled “Transplacental transfer of maternal antibody against SARS-CoV-2 and its influencing factors:a review”(Manuscript ID: vaccines-1716269). Those comments are all valuable and very helpful for revising and improving our paper, as well as the important guiding significance to our researches. We have studied comments carefully and made correction which we hope meet with approval. To make it easier for you to view the manuscript, we used the "Track Changes" function of MS Word and in red. The main corrections and responses to reviewers' comments in the paper are attached.

     Once again, thank you very much for your comments and suggestions.

Yours sincerely,

Shuang Liu

Corresponding author: Dingmei Zhang

Affiliation: Department of Epidemiology, School of Public Health, Sun Yat-sen University, Guangzhou, Guangdong 510080, China

Response to Reviewer 2 Comments

Point 1: 1-In the literature there are not many data on the subject and the focus of the review has a great and important social impact. The publication of the review by Marco A. P. Safadi on journal Am J Reprod Immunol. (April 22, 2022) reduces the novelty of the review which, however, remains important to enrich the literature; add the Safadi’s reference.

Response 1: Thank you very much for this helpful suggestion. We added the Safadi’s reference to the revised manuscript in lines 303-306, page8

Point 2: suggest to cited the Ching-Ju Shen’ article to strengthen the concept that Immunization of pregnant women is crucial to protect mothers and their offspring from infection.

Response 2: We sincerely appreciate your valuable comments. To reinforce the concept that maternal immunization is essential to protect mothers and their offspring from infection. We added the references of the revised manuscript in lines 306-307, page 8.
